# A Study of the Synergistic Interaction of Konjac Glucomannan/Curdlan Blend Systems under Alkaline Conditions

**DOI:** 10.3390/ma12213543

**Published:** 2019-10-29

**Authors:** Weijian Ye, Bowen Yan, Jie Pang, Daming Fan, Jianlian Huang, Wenguo Zhou, Xueqian Cheng, Hui Chen, Hao Zhang

**Affiliations:** 1Key Laboratory of Refrigeration and Conditioning Aquatic Products Processing, Ministry of Agriculture and Rural Affairs, Xiamen 361022, China; Yeweijianch@163.com (W.Y.); yanbowen@jiangnan.edu.cn (B.Y.); Zhouwenguo@anjoyfood.com (W.Z.); chengxueqianch@163.com (X.C.); chen15180451451@163.com (H.C.); 2State Key Laboratory of Food Science and Technology, Jiangnan University, Wuxi 214122, China; zhanghao@jiangnan.edu.cn; 3College of Food Science, Fujian Agriculture and Forestry University, Fuzhou 350002, China; pang3721941@163.com; 4School of Food Science and Technology, Jiangnan University, Wuxi 214122, China; 5Fujian Anjoy food Share Co. Ltd., Xiamen 361022, China

**Keywords:** konjac glucomannan, curdlan, synergistic interaction, viscosifying effect, thermal stability, alkaline conditions

## Abstract

To improve the gelation performance of konjac glucomannan (KGM) thermo-irreversible gel in the condition of alkaline, this study investigated the interactions between KGM and curdlan (CUD) in terms of the sol state and gelation process. The apparent viscosity, rheological properties during heating and cooling, thermodynamic properties, gelation properties and water holding capacity of KGM/CUD blend systems in an alkaline environment were studied using physical property testing instruments and methods. The results showed that the viscosity of the KGM/CUD blended solution was greater than the value calculated from the ideal mixing rules in the condition of alkaline (pH = 10.58). As the proportion of CUD in the system increased, the intersection of storage modulus (G’) and loss modulus (G”) shifted to low frequencies, the relaxation time gradually increased, and the degree of entanglement of molecular chains between these two components gradually increased. The addition of CUD helped decrease the gelation temperature of KGM, increased the gelation rate and inhibited the thinning phenomenon of KGM gels at low temperatures (2–20 °C). The addition of CUD increased the hardness and gel strength of KGM but did not significantly improve the water holding capacity of the KGM/CUD blend gel. The process of mixing KGM and CUD improved the thermal stability of the gel. In summary, KGM/CUD exhibited excellent compatibility under alkaline conditions, and the blend systems produced a “viscosifying effect”. KC8 and KC5 show better thermal stability, low temperature resistance and gel strength compared to KGM. This blended gel can be used as a structural support material to provide reference for the development of konjac bionic vegetarian products.

## 1. Introduction

Natural polysaccharides often possess thickening, gelling, adhesion, film forming, and biodegradability properties and are important raw materials widely used in the fields of food, chemical, pharmaceutical and biological etc. Due to the limited properties of single polysaccharides, their applications are subject to certain restrictions. Under certain conditions, some polysaccharides generate new properties through intermolecular or intramolecular interactions, thereby improving or enhancing the properties of single polysaccharides and broadening the applications. Since the 1970s, blended systems of natural polysaccharides and polysaccharides have been extensively studied to determine the interaction mechanisms of polysaccharides, changes in sol and gel properties, and their applications in different fields as functional materials. For example, hydrogen bonds exist between the molecules of pectin and konjac glucomannan (KGM), and this mixture has been combined with tea polyphenols to produce food packaging materials with biological activity [1]. The blend gel of KGM and gellan gum has been used as a fat substitute in frankfurter sausage to enhance its juicy texture [2,3]. Chitosan derivatives and alginate form interpenetrating network structures and have been used as pH-sensitive hydrogels for targeted drug release [4]. A composite hydrogel with excellent biocompatibility prepared from KGM and carboxymethyl chitosan has been used as a wound dressing for wound healing [5]. In addition, blended polysaccharides have been used in intelligent hydrogels [6,7], aerogels [8], soil improvement [9], food moisture retention [10,11], and biological materials [12,13,14,15]. Studies on the interactions between polysaccharides have important practical application value.

KGM is a natural, soluble and neutral polysaccharide extracted from the roots and tubers of *Amorphophallus konjac K. Koch*, which has a long history as a food source in the subtropical and tropical regions of Asia [16], and has been widely used in food processing, chemical engineering, agriculture, medicine and other fields [17,18,19,20]. Gelation is one of the most important properties of KGM, which can form thermo-reversible and thermo-irreversible gel under certain conditions [21]. Among them, the thermal-irreversible gel can be formed when the KMG in alkaline condition (pH control of 9–12) with heating (general temperature controlled at 85–100 °C), which has been used to make many kinds of bionic vegetarian food and traditional Chinese food [22], such as vegetarian abalone, vegetarian beef tripe, vegetarian sea cucumber, konjac tofu and konjac fans, and combine with surimi to improve the quality of surimi products [23], etc. However, there are some problems have limited its application when the KGM thermo-irreversible gel was used alone in alkaline condition, such as a high rate of syneresis, intolerance to cryopreservation, strong alkaline taste and dark colour [21]. Fortunately, studies have shown that blending KGM with natural polysaccharides such as carrageenan [24], xanthan gum [25] can improve the properties of KGM gel, which provides a reference direction for solving the above problems.

Curdlan (CUD) is a neutral microbial polysaccharide with a linear structure composed of D-glucose chains formed from β-1, 3 glucoside bonds [26] and insoluble in cold water, but soluble in alkaline solution and DMSO [27]. CUD possesses unique gel properties that are closely related to the heating temperature and forms thermoreversible and thermo-irreversible gels, suggesting it is a good gel material. It has been used as an immobilization carrier in chemical engineering, construction and other fields [28,29]. It has also been used as a biologically active substance in the medical field [30,31] with antitumor, antibacterial, drug delivery and other functions. Furthermore, it has been used as stabilizers, gels, and quality improvers in food industry, particularly in biomimetic food, meat products and rice and flour products [32,33]

Both KGM and CUD are neutral polysaccharides that are able to form thermoreversible gels and thermo-irreversible gels under certain conditions. At the same time, hydrogen bonds form between the blending film composed of the KGM and CUD [34], and the film has excellent moisture resistance and mechanical properties. However, under different solution conditions, the properties of the two colloids are altered to some extent. Alkaline conditions promote both KGM deacetylation and CUD dissolution. We speculate that the sol viscosity and gel properties of the blend gel may change following hydrogen bonding and other interactions. In this paper, blend systems of KGM and CUD with different ratios in alkaline water dispersion medium were studied. The viscosity, the gelling process of the blend systems, the thermal stability, the textural characteristics and the water holding capacity of the KGM/CUD blend gel were investigated to improve the quality of KGM thermo-irreversible gel in alkaline conditions and provide a reference for the application of KGM/CUD blend hydrogels.

## 2. Materials and Methods 

### 2.1. Materials

KGM (glucoglucan content of 92%) was purchased from Sanai Organic Konjac Development Co., Ltd., Zhaotong, Yunnan, China. CUD (purity 90%) was purchased from Mitsubishi Life Technologies, Tokyo, Japan. Sodium carbonate, silicone oil and citric acid were purchased from Sinopharm Chemical Reagent Co., Ltd. Shanghai, China.

### 2.2. Preparation of KGM/CUD Blend Sols 

(1) Preparation of KGM sol: A certain amount of KGM was dissolved in deionized water at a concentration of 1.0% (*w*/*w*) by stirring, and the mixture was continuously stirred for 1 h at 250 r/min in a 45 °C water bath.

(2) Preparation of CUD sol: A certain amount of CUD was dissolved in deionized water containing 0.1% sodium carbonate at a concentration of 1.0% (*w*/*w*) by stirring, and the mixture was continuously stirred for 1 h at 250 r/min in a 45 °C water bath.

(3) Preparation of KGM/CUD blend sols: The sols prepared in steps (1) and (2) were blended at KGM:CUD weight ratios of 10:0, 8:2, 5:5, 2:8, and 0:10 (labeled as KGM, KC8, KC5, KC2, and CUD). The pH value was measured with the S-2C pH meter from Shanghai Weiye Instrument Factory (Shanghai, China) and was adjusted to 10.58 by adding sodium carbonate while stirring and mixing.

(4) Preparation of KGM/CUD blend gels: The concentration ratio of the two polysaccharides was 2.0% (*w*/*w*). Blend sols were prepared separately as described in steps (1), (2) and (3), subjected to a vacuum of −0.08 MPa for 1 min, and then injected into a polyvinyl chloride (PVC) casing with a diameter of 2.3 cm. The gels were heated at a constant temperature of 90 °C in a water bath for 1.5 h and then cooled to room temperature.

### 2.3. Rheological Properties of the KGM/CUD Blend Sols 

Rheological properties were determined according to the method of Hu, Y. [35] with minor modifications. The rheological properties of the sols were measured using the MCR301 rheometer from Anton Paar Austria GmbH (Graz, Stmk, Austria). PP50 flat probes with a spacing of 1 mm were used in the following tests:

(1) Viscosity test: shear rate ranging from 1 s^−1^ to 1000 s^−1^ and test temperature of 25 °C;

(2) Frequency sweep test: strain of 1%, sweep frequency ranging from 0.1 rad/s to 100 rad/s, and test temperature of 25 °C; 

(3) Dynamic measurements of viscoelasticity during heating and cooling processes: the sample was placed in the sampler of the rheometer and sealed with silicone oil to prevent the evaporation of the liquid. The test conditions were a frequency of 1.0 Hz, strain of 1%, test temperatures of 25–100 °C (heating rate: 2 °C/min), 100–20 °C (cooling rate: 5 °C/min), and 20–2 °C (cooling at low temperature and rate is 0.1 °C/min).

### 2.4. Texture Properties of the KGM/CUD Blend Gels

Gel strength was determined according to the method of Jian W.J. [36] with minor modification. After equilibrated at room temperature (25 ± 1 °C) for 0.5 h, the cylindrical gel (diameter 2.3 cm × height 3.0 cm) was used for determination. A puncture was created using a TA.XT Plus texture analyzer from Stable Micro Systems Ltd. (Surrey, United Kingdom). A spherical probe with a diameter of 5 mm was used. The test speed was 1.0 mm/s, trigger force was 2 cN, and the puncture depth was 15 mm. The following formula was used to calculate the gel strength:(1)Gel strength (g∗cm)=rupture force (g)× rupture distance (cm) 

### 2.5. Determination of the Water Holding Capacity of KGM/CUD Blend Gels

The blend gel was prepared using the method described in Section 2.2, step (4). Using the method reported by Beatriz H. [37] with appropriate modifications, the gel was cut into cylindrical samples with a diameter of 5 mm and a height of 2 cm using a mold, placed in a 1.5 cm-diameter centrifuge tube with 0.15 g of cotton at the bottom, and centrifuged at 5000 r/min for 15 min at room temperature. After the water on the surface of the gel was dried, the gel mass before centrifugation (m_1_) and gel mass after centrifugation (m_2_) were measured, and the water holding capacity was calculated. Five parallel gel samples were measured for each condition and the average result was recorded. The following formula was used to calculate the water holding capacity:(2)$$\mathrm{Water}\;\mathrm{holding}\;\mathrm{capacity}\;\left(\%\right)=\frac{m_{1}-m_{2}}{m_{1}}\times 100\%$$

### 2.6. Differential Scanning Calorimetry (DSC) Analysis of KGM/CUD Blend Systems 

Using the method reported by Wu, C.H. [34], a DSC 200F3 differential calorimetry scanner from Netzsch (Shanghai) Mechanical Instrument Co., Ltd. (Shanghai, China) was employed to test the thermodynamic properties of the blend gel. The blend gel was prepared using the method described in Section 2.2, step (4). The gel samples were divided into two groups. One group was freeze-dried, and the freeze-dried samples (6.5–8.5 mg) were placed in a sealed aluminum crucible. The degradation and stability of the KGM/CUD blend system were investigated at temperatures ranging from 20 °C to 400 °C with a heating rate of 10 K/min. The other group of untreated gel samples (6.5–8.5 mg) was placed in a sealed aluminum crucible to investigate the water holding stability of the blend systems at temperatures ranging from 25 °C to 120 °C with a heating rate of 10 K/min. All tests described above were performed under nitrogen gas at a flow rate of 50 ml/min, and an empty sealed aluminum crucible was used as a reference.

### 2.7. Statistical Analysis

All measurements were performed in triplicate. Figures were created with Origin 9.0 software. Significant difference among values were calculated based on Tukey’s procedure at *p* < 0.05, using the software SPSS statistics (version 17.0, International Business Machines Corporation, Armonk, NY, USA).

## 3. Results and Discussion

### 3.1. Apparent Viscosity of KGM/CUD Blend Systems 

According to the ideal mixing rules [38], when KGM and CUD do not interact, the theoretical viscosity of the blend solution is calculated using the following equation: (3)$$\ln\eta_{c}=\phi_{1}\ln\eta_{1}+\left(1-\phi_{1}\right)\ln\eta_{2}$$

The term $\phi_{n}$ represents the proportion of the corresponding component, $\eta_{n}$ (*n* = 1, 2) represents the actual viscosity of a single solution of the corresponding component, and *η_c_* is the corresponding theoretical viscosity. When the actual viscosity of the blend solution is greater than the viscosity calculated according to the ideal mixing rules, a “viscosifying effect” occurs in the blend solution and the two components interact; otherwise, the opposite is true. Figure 1 shows the comparison of apparent actual viscosity and theoretical viscosity of KGM/CUD blend sol in an alkaline environment. The viscosities of KGM, CUD and KGM/CUD blend sol decreased as the shear frequency increased, exhibiting the shear-thinning phenomenon. The actual viscosities of KC8, KC5 and KC2 were greater than the corresponding theoretical viscosities, indicating that under alkaline conditions, a certain synergy existed between KGM and CUD, resulting in an increase in the viscosity of the sol. KGM contains a large amount of –OH, C=O and other groups. After dispersion in water, KGM molecules exhibit an extended helical structure [39]. Using O(6) and O(2) in the molecular chain as nodes, a large number of intermolecular and intramolecular hydrogen bonds with water molecules or KGM molecular chains are formed, and thus the entire system forms a hydrogen bond network structure, resulting in high viscosity [40]. Under alkaline conditions, KGM aggregation occurs [41]. At the same time, after the gradual dissolution of the CUD granular powder in water, the molecular chain of CUD expands and part of the triple helix dissociates into a single helix [27], forming a more uniform solution. Therefore, CUD may be physically entangled with the molecular chains of KGM to form molecular aggregates, thus increasing the viscosity of KGM/CUD blend systems. 

### 3.2. Viscoelasticity of the KGM/CUD Blend Systems during the Dynamic Frequency Sweep

Frequency scanning is mainly used to detect the structure and relaxation properties of materials. As shown in Figure 2, under alkaline conditions (pH = 10.58), the curves of KGM and CUD solutions have typical characteristics of dilute and concentrated polymer solutions, respectively [42]. At the low-frequency scanning range (<15 rad/s), G” was always larger than G’ for KGM and KC8. A potential explanation is that with the decrease of the applied frequency of strain, the original entangled polysaccharide molecular chain is gradually untied, and the energy is lost through the flow of viscous molecules, which is then comprehensively manifested as a viscous character. Within the scope of the test, G’ and G” increased with increasing frequency, potentially because the molecular chains are not fully extended or oriented within a short period of time, leading to the transient existence of a network structure between molecular chains. The increase in G’ and G” of CUD with increasing frequency was not particularly obvious, and G’ was always larger than G”. After swelling in a water bath at 45 °C, a weak CUD gel gradually forms, and the comprehensive manifestation is greater elastic behavior and stability than KGM. As the proportion of the CUD component in the system increases, the intersection of G’ and G” shifted to the low frequency, the relaxation time gradually increased, and the frequency of the transformation of the two types of modulus for KGM and KC8 were 15.8 rad/s and 10 rad/s, respectively. Based on these results, the entanglement of the molecular chains of these two components gradually increased and the stability of the system improved.

The ratio of G” and G’ is tanδ. When tanδ is greater than 1, the viscosity dominates; when tan δ < 1, elasticity dominates. As shown in Figure 3, an increase in the CUD content caused the blend systems to gradually exhibit solid-state characteristics, and the curve tended to be flat. When the ratio of KGM and CUD was 5:5, the system was fully characterized by a solid state, indicating that the structure of the blend was stable and maintained good elasticity.

### 3.3. Viscoelasticity of KGM/CUD Blend Systems during the Heating Process

In Figure 4, the G” and G’ of KGM initially exhibited a gradual decrease as the temperature increased under alkaline conditions (pH = 10.58). In the range of 91–99 °C, the rate of increase in G’ was significantly greater than G”. When the temperature reached 97 °C, G’ was larger than G”, and KGM exhibited the gelation phenomenon. Under alkaline conditions during heating, KGM is deacetylated, and the KGM molecular chain becomes naked. Following the formation of hydrogen bonds and hydrophobic interactions, the molecular chains intertwine to form a thermo-irreversible gel with a network structure [43]. Because CUD was dissolved at pH = 10.58 and 45 °C for 1 h, it resembled a weak gel when tested at 25 °C; at 25–40 °C, G’ was greater than G”, showing elastic gel behavior. As the temperature continued to increase, G’ and G” began to decrease and returned to the sol state at 53–57 °C, indicating that the gel that formed at 25–53 °C was a thermoreversible gel. When the temperature continued to increase (>57 °C), the spiral spacing of the CUD molecules decreased, the force increased, G’ was much larger than G”, the elastic properties were more obvious, and CUD formed a thermo-irreversible gel.

In Figure 5a, during the heating process, the temperatures at which G’ and G” intersected in KC8, KC5, and KC2 were 75 °C, 76 °C, and 71 °C, respectively, and all were in the gelation temperature range of KGM and CUD, indicating that the introduction of CUD helped reduce the gelation temperature of KGM. In Figure 5b, at the late stage of the heating process (85–99 °C), G’ was in the following order: KC5 > KC8 > KC2 > KGM > CUD, and with the addition of CUD, the slope of the curve increased, particularly for KC8 and KC5, indicating that CUD increased the gel formation rate of the blend. As the temperature increased, G’ of KC8 and KC5 significantly increased, indicating that CUD improved the gel strength and high temperature resistance of the blend systems, which was consistent with the results of texture and DSC analysis.

### 3.4. Viscoelasticity of KGM/CUD Blend Systems during the Cooling Process

The gelation of KGM is closely related to the concentration, heating temperature, time, and pH value. As shown in Figure 6a, G’ gradually decreased as the temperature decreased, while G” did not exhibit substantial changes. When the temperature reached 12.6 °C, the gel and sol transition point began to appear. When the temperature further decreased, G” was greater than G’, indicating that the gel was transformed into a sol; namely, the KGM gel was relatively unstable at low temperatures (2–20 °C). This may be due to the low degree of deacetylation of KGM, which is easier to transform from gel to sol at low temperature. During the cooling process, both G’ and G” of CUD gradually increased and G’ was far greater than G”, indicating a relatively stable gel state. As the temperature increases, hydrogen bonds in the system are destroyed, CUD forms a thermo-irreversible gel through hydrophobic interactions, hydrogen bonds are regenerated after the temperature decreases [44], and the gel strength of the system is further improved. Therefore, the CUD gel exhibited high stability at low temperature. As shown in Figure 6b, G’ was greater than G” for KC8, KC5, and KC2 throughout the low temperature range, and the blend systems exhibited elastic behavior. In this case, the addition of CUD inhibited the thinning phenomenon of the KGM gel at low temperatures (2–15 °C), and the blend gel was more stable at low temperatures.

### 3.5. Texture Properties and Water Holding Capacity of KGM/CUD Blend Systems

Figure 7 shows the variations in the rupture force, rupture distance, gel strength and water holding capacity of the KGM/CUD blend systems produced at different ratios. The rupture distance of CUD was relatively large and CUD was more elastic. As the CUD content increased, the water holding capacity and rupture distance of the blend systems did not exhibit substantial changes, while the rupture force and gel strength increased significantly, indicating that the addition of CUD improved the rupture force and gel strength of KGM.

### 3.6. DSC Analysis of KGM/CUD Blend Systems 

As shown in Figure 8, a broad absorption peak was observed for KGM, CUD, KC8, KC5 and KC2 at a wide range of temperatures, from 75 °C to 90 °C, due to the heat of vaporization of water in each sample. The exothermic peak at approximately 328.0 °C for KGM is caused by thermal degradation, the exothermic peak at 205.9 °C for CUD is also caused by thermal degradation, and the thermal degradation temperatures of the two components are quite different. When KGM and CUD were blended at different ratios, the temperature of the exothermic peak corresponding to degradation shifted towards a higher temperature compared with CUD. The temperatures of KC2, KC5, and KC8 were 220.6 °C, 221.7 °C, and 216.6 °C, respectively, and all blends produced a single peak, indicating good compatibility between the two gels under alkaline conditions.

In Figure 9, an absorption peak was observed for KGM, CUD, KC8, KC5 and KC2 gels at temperatures ranging from 100 °C to 114 °C, due to the heat of vaporization of water. The smaller the temperature corresponding to the absorption peak, the lower the binding capacity of the gel for water. A broad endothermic peak for the KGM gel was observed at 103.7 °C with an area of 0.55 J/g, whereas the endothermic peak of the CUD gel was observed 108.0 °C with an area of 0.65 J/g, indicating that the ability of the CUD gel to bind water molecules was greater than the KGM gel under the same conditions. When the CUD content in the gel system increased, the temperature corresponding to the endothermic peak first increased and then decreased: KC8 was 110.4 °C, KC5 was 109.1 °C, and KC2 was 108.2 °C. Thus, the process of blending KGM and CUD improved the thermal stability of the system to an even greater extent than the single systems composed of KGM and CUD.

In conclusion, the possible action mechanism of KGM/CUD was analyzed. The formula for KGM is (C_6_H_10_O_5_)_n_, and its basic structure (as shown in Figure 10a) is considered to be formed by polymerizing D-glucopyranose and D-mannopyranoside through β-1,4 glucosidic bond [45]. There are branches which are linked by β-1, 3 glycosidic bonds at the C-3 position of some saccharide residues. There are acetyl groups bonded by ester bonds at the C-6 position of the short chain of the side chain, and about every one of the 19 sugar residues has one acetyl group combined with a main chain or a branched sugar residue [46,47,48]. Under alkaline conditions with heating, the acetyl group was removed from KGM, the spiral structure was destroyed, and intermolecular and intramolecular hydrogen bonds between the hydroxyl group and the water molecule on the glycosyl group formed. At the same time, in the presence of hydrophobic forces, the molecular chains are interwoven, thereby generating partial structural crystallization and forming a gel with the crystal as a node [49,50]. In addition, the higher degree of deacetylation, temperature and alkali concentration, the better interaction between molecular chains and hydrogen bond was formed [51,52]. The formula for CUD is (C_6_H_10_O_5_)_n_ and is composed by D-glucose through β-(1,3)-D-glucoside bond. It is a non-branched linear polysaccharide [26] (as shown in Figure 10b). CUD can form two types of gels [53]. The CUD aqueous suspension or solution is heated to about 55–65 °C, and then cooling down to room temperature, a thermoreversible low-set gel is formed. In addition, when the CUD aqueous suspension is directly heated to above 80 °C, it can form a thermo-irreversible high-set gel, and the gel strength is further increased after cooling. The gel mechanism can be explained as follows [44]. When the temperature of the CUD is higher than 55 °C, the hydrogen bond between the molecular chains of CUD breaks and the hydrophobic interaction becomes dominant. In addition, under the action of hydrophobic forces, the single-helix and incomplete single-helix molecular chains of CUD are gradually transformed into triple helical molecular chains and the mutual force is enhanced to further form a gel network structure induced by it. After cooling, it produces another gel structure dominated by hydrogen bond. Previous studies have shown that the cleavage of hydrogen bonds, hydrophobic interactions and the reconstruction of hydrogen bonds are the basic steps in the formation of CUD thermo-irreversible gels. 

Polysaccharide hybrid gels are commonly divided into fully interpenetrating network hydrogels and semi-interpenetrating network hydrogels [54]. The fully interpenetrating network hydrogels are obtained by cross-linking the two crosslinked polymers; semi-interpenetrating network hydrogel means that of the two polymers constituting the interpenetrating network structure, but only one polymer is crosslinked and the other polymer is linear non-crosslinked. As the rheological test in this study, it is known that both KGM and CUD and their blending system can form gels under alkaline conditions. In addition, DSC research shows that the two have good compatibility. Combined with Wu, C.H. [34] studies, we speculate that the intermolecular and intermolecular entanglement of the two molecular chains may form the interpenetrating network structure due to the simultaneous existence of intramolecular, intramolecular hydrogen bonds and hydrophobic interactions in the KGM and CUD blends, which resulted that the G’ and gel strength of KC8, KC5 and KC2 increased compared with KGM. 

## 4. Conclusions

Under alkaline conditions (pH = 10.58), KGM/CUD blend systems produced a “viscosifying effect”. The addition of CUD decreased the gelation temperature of KGM, increased the gelation rate and inhibited the thinning phenomenon of the KGM gel at low temperatures (2–20 °C). A low proportion of CUD improved the hardness and gel strength of the KGM gel but did not significantly improve the water holding capacity of the blend gel. The compatibility, gel thermal stability and low temperature resistance of the KGM/CUD blend systems were good, further KC8 and KC5 exhibited better thermal stability, low temperature resistance and gel strength compared with KGM.

## Figures and Tables

**Figure 1 materials-12-03543-f001:**
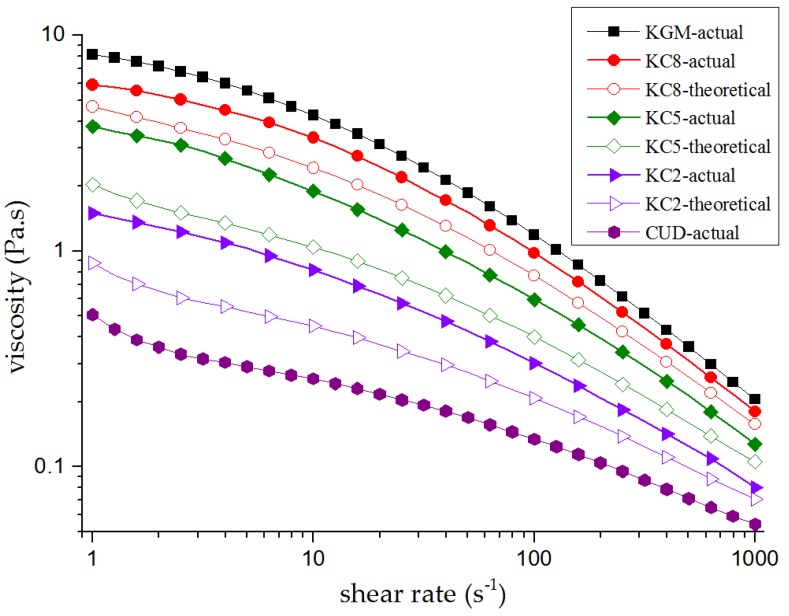
Relationships between the shear rate and both theoretical and actual viscosities of konjac glucomannan (KGM)/curdlan (CUD) blend systems under alkaline conditions (pH = 10.58).

**Figure 2 materials-12-03543-f002:**
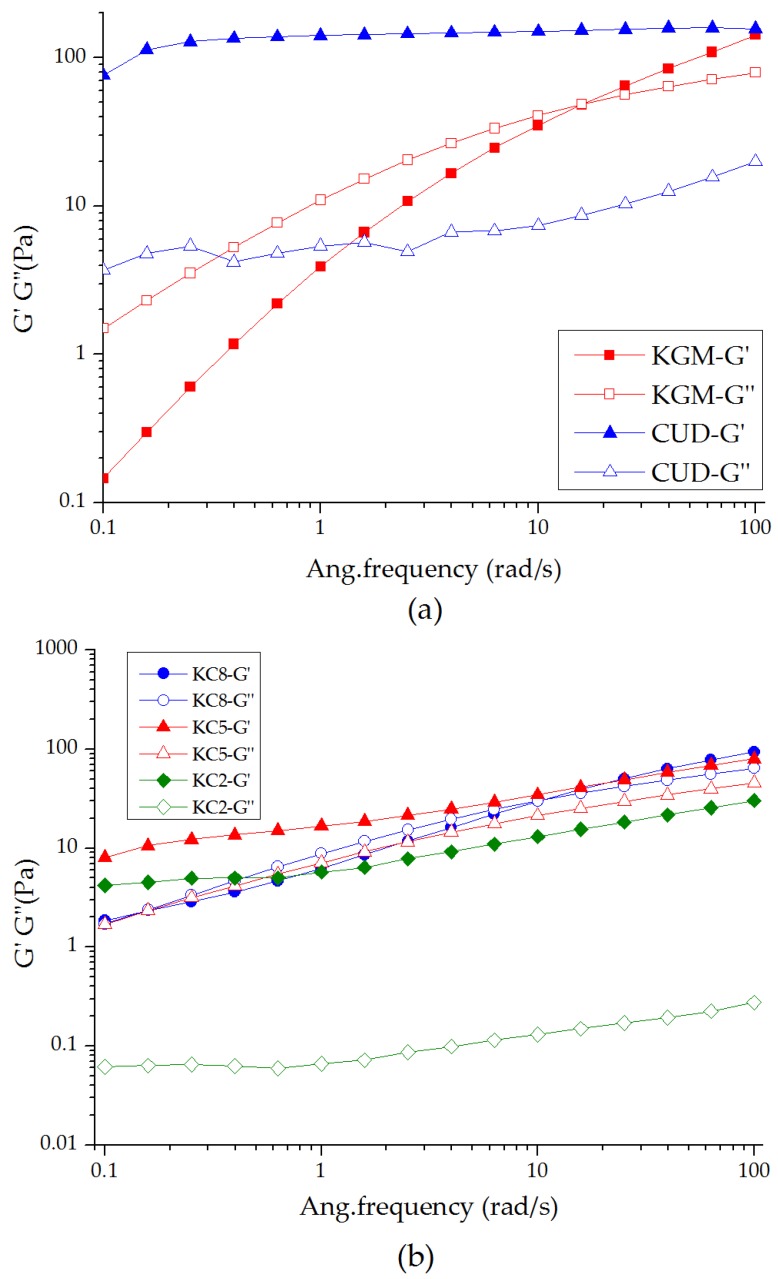
Curves showing the variations in G’ and G” of (**a**) KGM and CUD and (**b**) KGM/CUD blend systems with different mixture ratio at 25 °C at different angular frequencies.

**Figure 3 materials-12-03543-f003:**
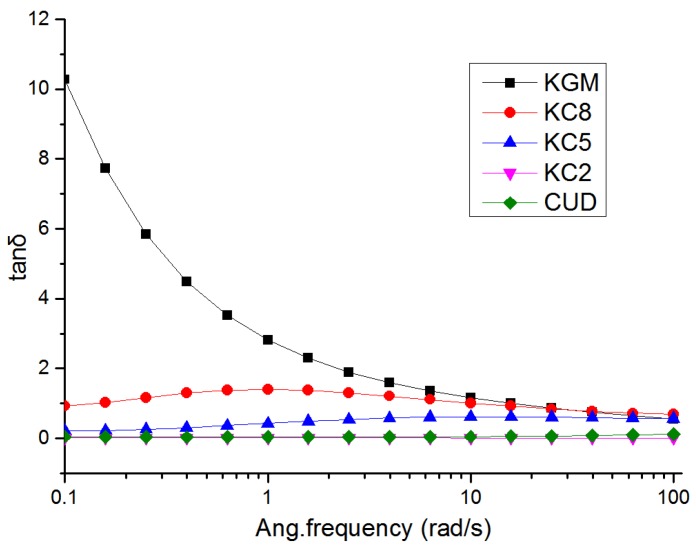
Curve showing the variation in tanδ for the KGM/CUD blend system at different angular frequencies.

**Figure 4 materials-12-03543-f004:**
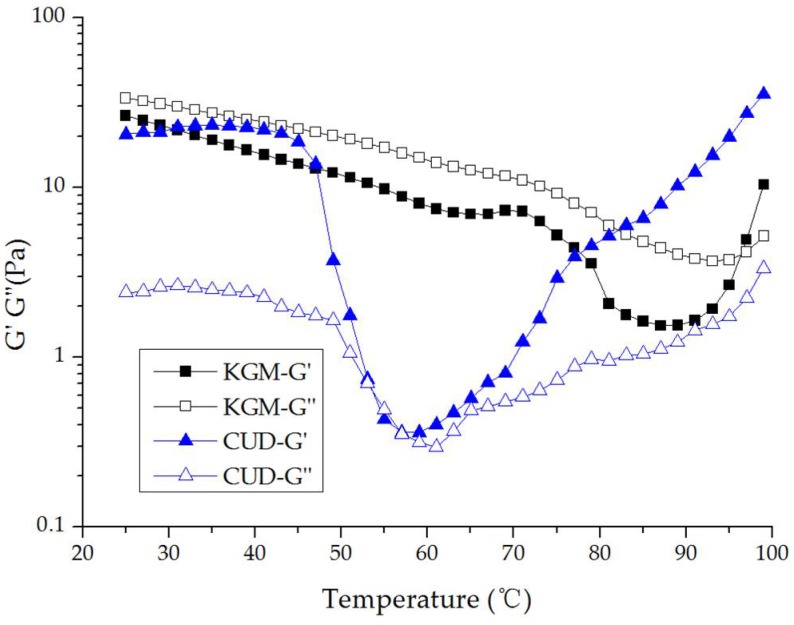
Curves showing the variation in G’ and G” for KGM and CUD with increasing temperature during the heating process.

**Figure 5 materials-12-03543-f005:**
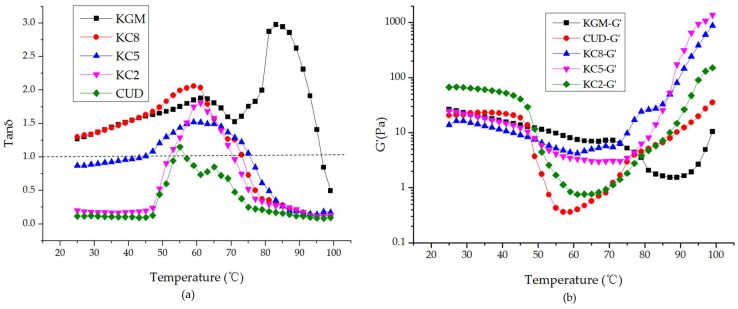
Curves showing the variation in (**a**) tanδ and (**b**) G’ for the KGM/CUD blend system with the increasing of temperature during the heating process.

**Figure 6 materials-12-03543-f006:**
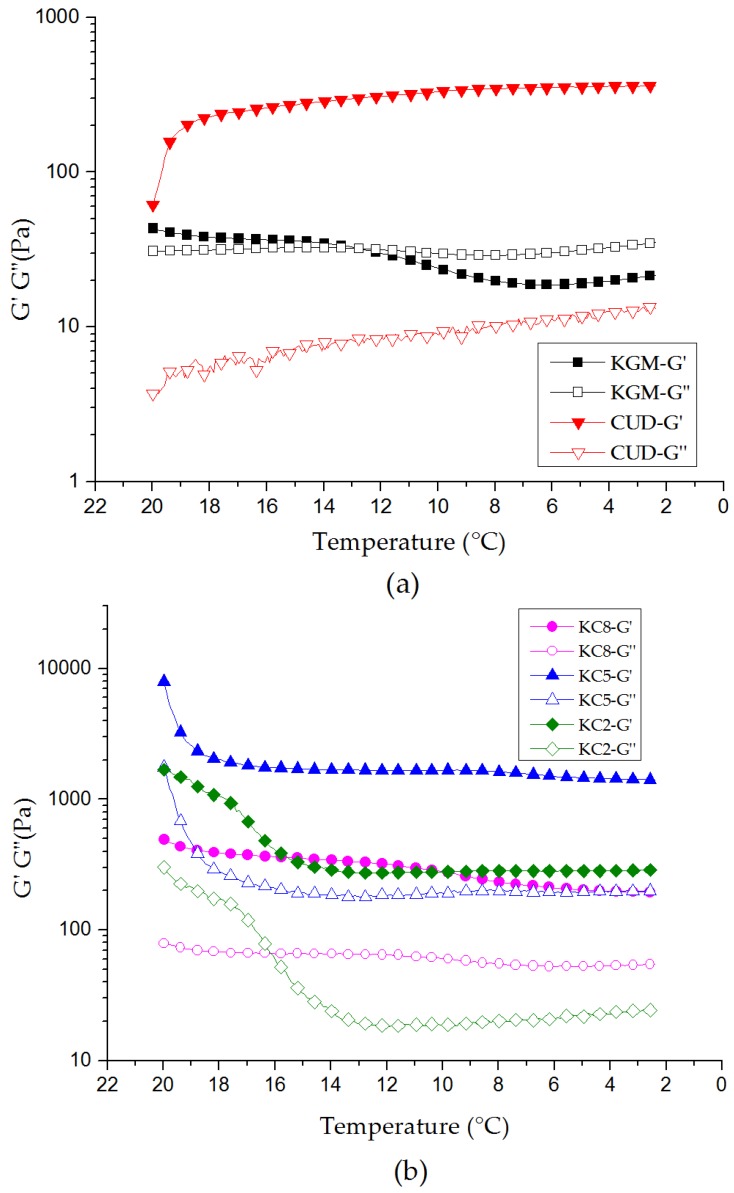
Curves showing the variations in G’ and G” of (**a**) KGM and CUD and (**b**) KGM/CUD blend systems with decreasing temperature during the cooling process (20–2 °C).

**Figure 7 materials-12-03543-f007:**
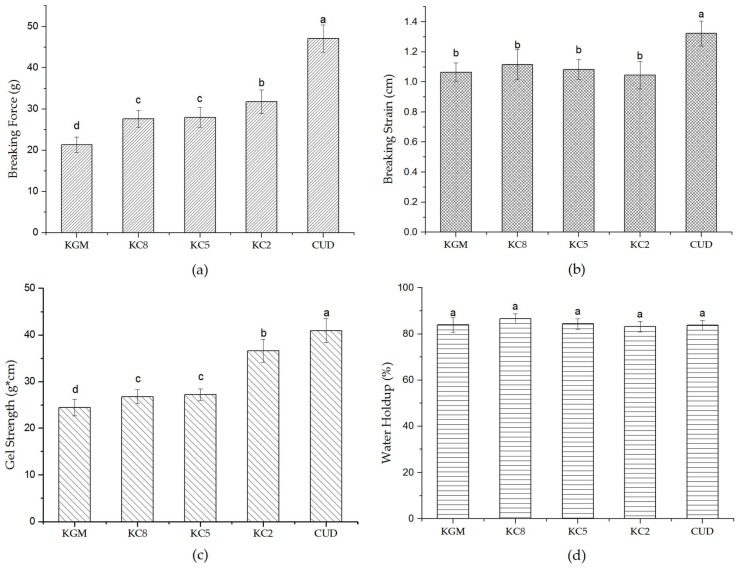
Curves showing the variations in the (**a**) breaking force, (**b**) breaking strain, (**c**) gel strength and (**d**) water holding capacity of the KGM/CUD blend systems. a, b, c, d letters indicate a significant difference (*p* < 0.05).

**Figure 8 materials-12-03543-f008:**
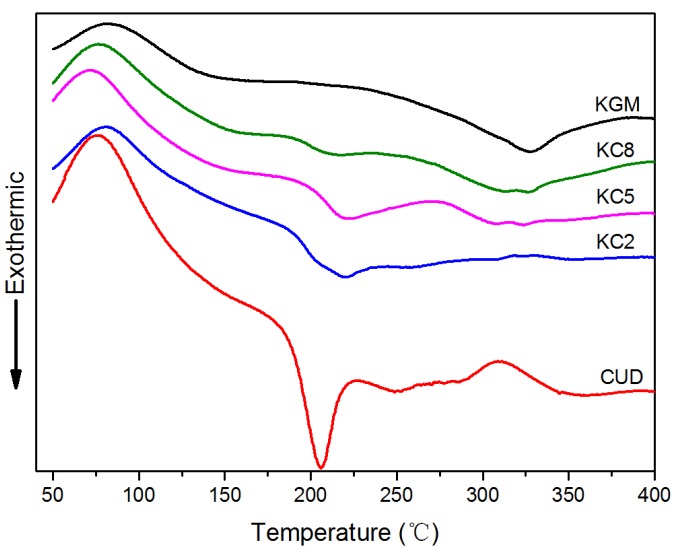
Differential scanning calorimetry (DSC) profiles of the freeze-dried KGM/CUD blend systems at 20–400 °C.

**Figure 9 materials-12-03543-f009:**
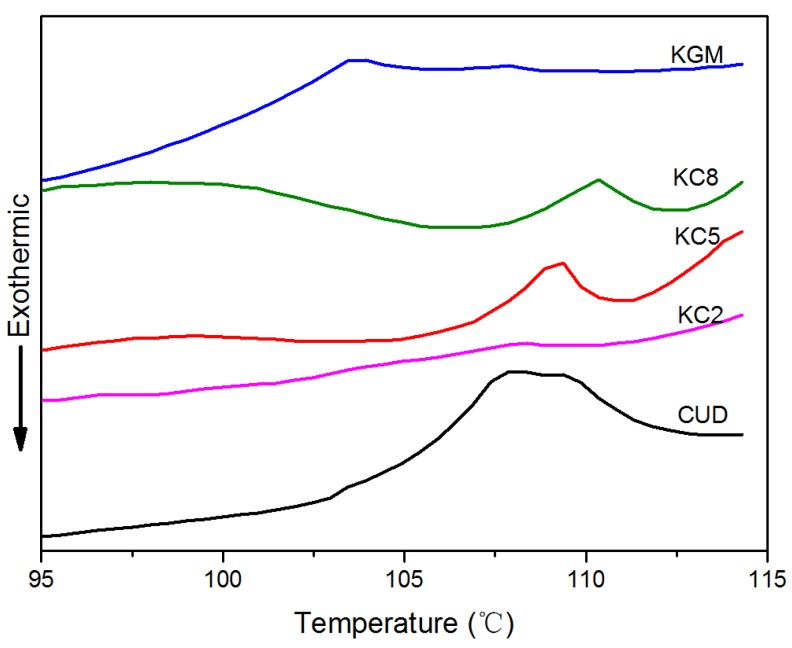
DSC profiles of KGM/CUD blend hydrogels at 95–115 °C.

**Figure 10 materials-12-03543-f010:**
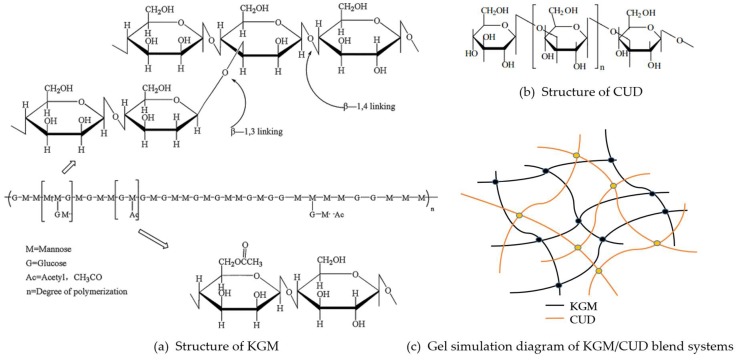
Structure of (**a**) KGM and (**b**) CUD and (**c**) gel simulation diagram of KGM/CUD blend systems.

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
