# Peer review of "A Study of the Synergistic Interaction of Konjac Glucomannan/Curdlan Blend Systems under Alkaline Conditions"

_materials, 2019, doi:10.3390/ma12213543_

Round 1
Reviewer 1 Report
The manuscript materials-608590 demonstrates the development of hydrophilic polymeric materials and in my opinion can be published in Materials journal.
However, I think that the manuscript can be improved for reading. This will increase interest for a wider range of readers. Here are some suggestions:
Abstract should show the essence of the article, in my opinion, and not be a summary of what the authors did.Abstract must be rewritten with correction of contradiction.For example, the pH value (line 21) is not consistent with the intended practical use of the materials (lines 31-33).In the abstract, it is necessary to indicate which ratio of the components is most effective and for what purpose. The purpose of the work is poorly marked, the choice of research objects is poorly justified, especially against the background of information - line 62-64. It is necessary to give the chemical formulas of KGM and of CUD and additional analytical data characterizing their structure and the structure of mixed system. It is also necessary to confirm the authors' reasoning about the interaction of components, since only one research method was used in this work. What is unique about pH 10.58?If the pH is 10.00, then how will the system behave? A single pH value significantly levels the significance of the results obtained, since it cannot be applied under other conditions. Figures 2, 3, and 5: It is inconvenient to compare data shown on different graphs. The values of G` and G`` must be given on separate plots for all compounds. Line 232-235: It is necessary to provide experimental evidence of the assumption made. The conclusion should be rewritten. It is not clear what specific result is useful to the reader.Author Response
Please see the attachment.

Reviewer 2 Report
in figures 1, 2, and 3 please report the standard deviations and the correlation coefficients
Please at line 253-254 clarify the Chinese term
figure 6 clarify if the error bars are SEM or Std. Dev
Round 2
Reviewer 1 Report
The second version of the manuscript materials-608590 can be published in present form.